# Beyond the Final Layer: Hierarchical Query Fusion Transformer with Agent-Interpolation Initialization for 3D Instance Segmentation

## Abstract

3D instance segmentation aims to predict a set of object instances in a scene and represent them as binary foreground masks with corresponding semantic labels. Currently, transformer-based methods are gaining increasing attention due to their elegant pipelines, reduced manual selection of geometric properties, and superior performance. However, transformer-based methods fail to simultaneously maintain strong position and content information during query initialization. Additionally, due to supervision at each decoder layer, there exists a phenomenon of object disappearance with the deepening of layers. To overcome these hurdles, we introduce Beyond the Final Layer: Hierarchical Query Fusion Transformer with Agent-Interpolation Initialization for 3D Instance Segmentation (BFL). Specifically, an Agent-Interpolation Initialization Module is designed to generate resilient queries capable of achieving a balance between foreground coverage and content learning. Additionally, a Hierarchical Query Fusion Decoder is designed to retain low overlap queries, mitigating the decrease in recall with the deepening of layers. Extensive experiments on ScanNetV2, ScanNet200, ScanNet++ and S3DIS datasets demonstrate the superior performance of BFL.

## 1 Introduction

Indoor instance segmentation is one of the fundamental tasks in 3D scene understanding, aiming to predict masks and categories for each foreground object. With the increasing popularity of AR/VR Park et al. (2020); Manni et al. (2021), 3D indoor scanning Lehtola et al. (2017); Halber et al. (2019), and autonomous driving Neven et al. (2018); Yurtsever et al. (2020), 3D instance segmentation has become a pivotal technology enabling scene understanding. However, the complexity of scenes and the diversity of object categories pose significant challenges to 3D instance segmentation.

To address the aforementioned challenges, a series of 3D instance segmentation methods Yi et al. (2019); Hou et al. (2019); Yang et al. (2019); Engelmann et al. (2020); Liu et al. (2020); Chen et al. (2021b); Liang et al. (2021); Vu et al. (2022); Schult et al. (2022); Sun et al. (2023); Lu et al. (2023); Lai et al. (2023) have been proposed. Generally, these methods can be categorized into three groups: proposal-based Yi et al. (2019); Hou et al. (2019); Yang et al. (2019), grouping-based Engelmann et al. (2020); Liu et al. (2020); Jiang et al. (2020b); Chen et al. (2021b); Liang et al. (2021); Vu et al. (2022), and transformer-based Schult et al. (2022); Sun et al. (2023); Lu et al. (2023); Lai et al. (2023). Proposal-based methods adopt a top-down approach, where they first extract 3D bounding boxes and then utilize a mask learning branch to predict the object mask within each box. Grouping-based methods initially generate predictions for each point (e.g., semantic categories and geometric offsets) and then generate instance proposals. Recently, transformer-based methods have attracted researchers' attention due to their elegant pipelines, reduced manual selection of geometric properties, and superior performance. These methods typically initialize a fixed number of object queries, which are then fed into the decoder to aggregate scene features. After the feature aggregation of each decoder layer, the queries output instance predictions, with each layer's predictions supervised by the ground truth. We refer to this design as per-layer auxiliary loss. The predictions from the final layer are used as the final output. In this process, query initialization plays a crucial role. Current transformer-based methods propose various designs for query initialization, mainly categorized into FPS-based (farthest point sampling) Schult et al. (2022); Lu et al. (2023) and learnable-based Sun et al. (2023); Lai et al. (2023) approaches. Furthermore, inspired by 2D instance segmentation Cheng et al. (2022); Li et al. (2023); Jain et al. (2023), the design of per-layer auxiliary loss has significantly improved the

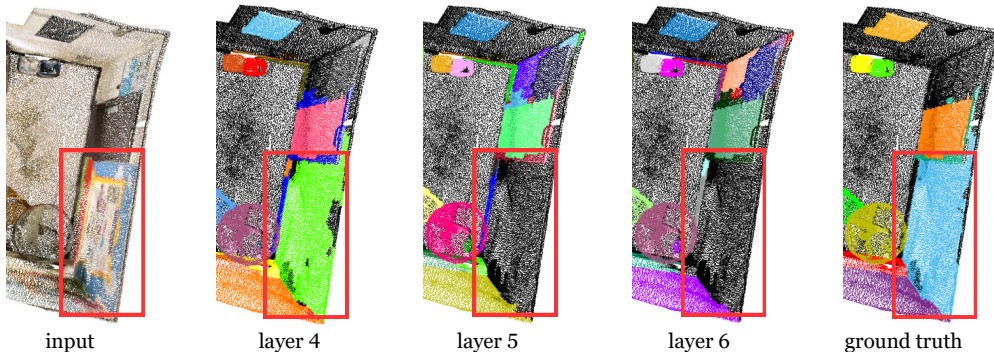

Figure 1: **The phenomenon of *Object Disappearance* with the deepening of layers.**

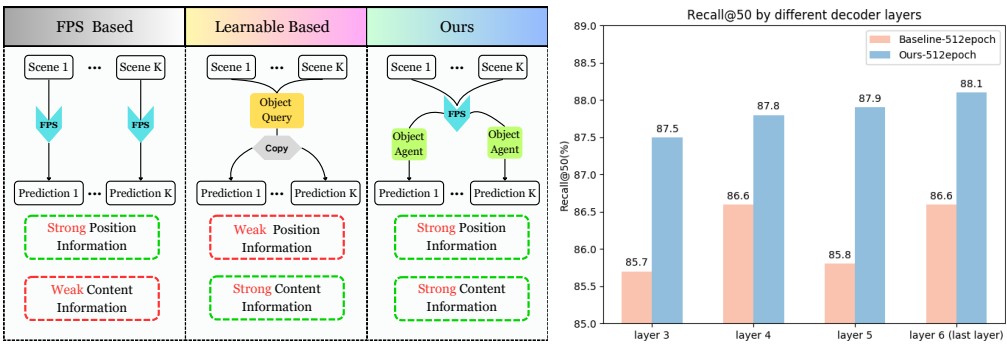

**(a) Agent-Interpolation Initialization Module**    **(b) Recall Difference**

Figure 2: **(a) The comparison of different query initialization methods.** The FPS-based methods conduct farthest point sampling separately for each scene, placing more emphasis on positional information but lacking in aggregating content information. The learnable-based methods initialize a fixed number of queries for aggregating content information across all scenes, which is prone to empty sampling, thereby compromising foreground coverage. Our method leverages the advantages of both approaches to achieve a balanced and comprehensive solution. **(b) The recall difference.** The recall of the baseline shows instability during the iterative optimization process across layers, whereas our method, with the assistance of the Hierarchical Query Fusion Decoder, demonstrates a steady improvement in recall across each layer.

training effectiveness of 3D instance segmentation. However, we observe a phenomenon of ***Object Disappearance***, where predictions for certain objects vanish as the deepening of layers. As shown in Figure 1, where the object "picture" obtained from the prediction at layer 4 disappears in layer 5 and layer 6. This is reflected in a decrease in recall in the quantized results, as shown in Figure 2 (b), contradicting the intuition that more interactions between features lead to better results.

Based on the discussion above, we have identified two challenges that need to be addressed: 1) *How to better initialize queries?* As illustrated in Figure 2 (a), current transformer-based methods Schult et al. (2022); Sun et al. (2023); Lu et al. (2023); Lai et al. (2023) can mainly be categorized into FPS-based Schult et al. (2022); Lu et al. (2023) and learnable-based approaches Sun et al. (2023); Lai et al. (2023). Mask3D Schult et al. (2022) and QueryFormer Lu et al. (2023) utilize FPS to obtain the initialization distribution of queries, which can more likely distribute candidates to the region where objects are located, thus reducing the empty sampling rate. However, these FPS-based approaches fail to learn content embedding across scenes effectively for feature aggregation. On the other hand, SPFormer Sun et al. (2023) and Maft Lai et al. (2023) employ learnable queries, which can update and learn across multiple scenes in the dataset. Nevertheless, the empty sampling rate is higher, leading to a decrease in model recall. Therefore, balancing the sampling positions of candidates and learning content embedding effectively is crucial for initializing queries. 2) *How to mitigate the issue of inter-layer recall decline?* During the decoding phase, due to the existence of auxiliary loss, the predictions of each decoder layer are supervised by ground truth. For instances that are difficult to predict, such as pictures, bookshelfs, the quality of the mask corresponding to the matched query is poor. Consequently, the mask attention Schult et al. (2022); Sun et al. (2023) focuses on a large amount of noisy features, causing the optimization direction of the query to be unstable, and there is a possibility of further deterioration in mask quality. Moreover, for other unmatched queries, due to the

lack of supervision signal, the optimization direction is even more random. Predicting better quality for such difficult-to-predict instances is therefore more challenging. As a result, the mask of instance "picture" in Figure 1 is lost by layer 5, and recall decreases. To address this issue, one intuitive idea is to concatenate the outputs of each layer's predictions during model inference, and then filter out duplicate predictions through non-maximum suppression (NMS) Neubeck & Van Gool (2006). However, since it is challenging to select suitable hyperparameters and lacks accurate confidence scores, NMS often cannot filter out lower-quality duplicate masks while retaining non-repetitive instance masks. Therefore, an end-to-end, automated design is needed to ensure that inter-layer recall does not decrease.

To achieve the aforementioned objectives, we propose BFL. To better initialize queries, we introduce the Agent-Interpolation Initialization Module (AI2M), where we initialize a set of agents comprising two corresponding queries: position queries and content queries. Subsequently, we perform FPS on the scene point cloud and interpolate the agents' content queries to obtain the sampled points' content queries based on their positions and the positions of the position queries. This approach ensures high foreground coverage of initial queries, avoiding empty sampling, and learns content information across scenes through interpolation, thereby effectively aggregating object features. To mitigate the issue of inter-layer recall decline, we propose the Hierarchical Query Fusion Decoder (HQFD). Specifically, we compute the Intersection over Union (IoU) between predicted instance masks from the ($l$-1)-th layer and the $l$-th layer. Queries from the ($l$-1)-th layer, showing low overlap (*i.e.*, corresponding masks having low IoU values with all masks from the $l$-th layer), are merged with queries from the $l$-th layer and collectively fed into the ($l$+1)-th layer. This method effectively retains queries with low overlap that aid in recall, mitigating the decrease in recall caused by unstable optimization directions. It's worth noting that the number of queries with low overlap is limited, so the extra queries added at each layer are few. This results in minimal impact on computational load, with a 7.8% increase in runtime.

In conclusion, our main contributions are outlined as follows:

(i) We introduce a novel 3D instance segmentation method called BFL.

(ii) We introduce a new query initialization method termed the Agent-Interpolation Initialization Module. This module integrates FPS with learnable queries to produce queries that can adeptly balance foreground coverage and content learning. It proves to be tailored for navigating complex environments.

(iii) We design the Hierarchical Query Fusion Decoder to retain low overlap queries, mitigating the decrease in recall with the deepening of decoder layers.

(iv) Extensive experiments conducted on ScanNetV2 Dai et al. (2017), ScanNet200 Rozenberszki et al. (2022), ScanNet++ Yeshwanth et al. (2023), and S3DIS Armeni et al. (2016) datasets show that BFL can surpass state-of-the-art transformer-based 3D instance segmentation methods.

## 2 RELATED WORK

In this section, we briefly overview related works on 3D instance segmentation, including proposal-based methods Yi et al. (2019); Hou et al. (2019); Yang et al. (2019), grouping-based methods Engelmann et al. (2020); Liu et al. (2020); Wang et al. (2018; 2019); Lahoud et al. (2019); Jiang et al. (2020b); Engelmann et al. (2020); Han et al. (2020); Jiang et al. (2020b;a); Chen et al. (2021b); Liang et al. (2021); Vu et al. (2022), and instance segmentation with transformer Cheng et al. (2021; 2022); Schult et al. (2022); Sun et al. (2023); Lu et al. (2023); Lai et al. (2023).

**Proposal-based Methods.** Existing proposal-based methods are heavily influenced by the success of Mask R-CNN He et al. (2017) for 2D instance segmentation. GSPN Yi et al. (2019) adopts an analysis-by-synthesis strategy to generate high-quality 3D proposals, refined by a region-based PointNet Qi et al. (2017a). 3D-BoNet Yang et al. (2019) employs PointNet++Qi et al. (2017b) for feature extraction from point clouds and applies Hungarian MatchingKuhn (1955) to generate 3D bounding boxes. These methods set high expectations for proposal quality.

**Grouping-based Methods.** Grouping-based methods make per-point predictions, such as semantic categories and geometric offsets, then group points into instances. PointGroup Jiang et al. (2020b) segments objects on original and offset-shifted point clouds and employs ScoreNet for instance score prediction. SSTNet Liang et al. (2021) constructs a tree network from pre-computed superpoints

Figure 3: **The overall framework of our method BFL.** The Agent-Interpolation Initialization Module is meticulously crafted to synergize the strengths of FPS and learnable queries, producing object queries better suited for complex and dynamic environments. The Hierarchical Query Fusion Decoder is utilized to retain low overlap queries that aid in recall rate.

and splits non-similar nodes to obtain object instances. SoftGroup Vu et al. (2022) groups based on soft semantic scores instead of hard semantic predictions and refines proposals to enhance positive samples while suppressing negatives. However, grouping-based methods require manual selection of geometric properties and parameter adjustments, which can be challenging in complex and dynamic point cloud scenes.

**Instance Segmentation with Transformer.** Transformer Vaswani et al. (2017) has been widely applied in computer vision tasks such as image classification Dosovitskiy et al. (2020); Chen et al. (2021a), object detection Carion et al. (2020); Ding et al. (2019); Wang et al. (2023), and segmentation Zheng et al. (2021); Cheng et al. (2021; 2022); Lu et al. (2024) due to the self-attention mechanism, which models long-range dependencies. Recently, DETR Carion et al. (2020) has been proposed as a new paradigm using object queries for object detection in images. Building on the set prediction mechanism introduced by DETR, Mask2Former Cheng et al. (2022) employs mask attention to impose semantic priors, thereby accelerating training for segmentation tasks. The success of transformer has also become prominent in 3D instance segmentation. Following Mask2Former, each object instance is represented as an instance query, with query features learned through a vanilla transformer decoder, and the output from the final layer serving as the final prediction. Mask3D Schult et al. (2022) and SPFormer Sun et al. (2023) are the first works to utilize the transformer framework for 3D instance segmentation. They respectively employ FPS and learnable queries as query initialization. QueryFormer Lu et al. (2023) and Maft Lai et al. (2023) are improvements upon Mask3D and SPFormer, but still utilize FPS and learnable queries for query initialization. Our approach combines FPS and learnable queries, employing the Agent-Interpolation Initialization Module to produce object queries better suited for complex and dynamic environments. Additionally, we utilize the Hierarchical Query Fusion decoder to retain low overlap queries that aid in recall rate.

## 3 METHOD

### 3.1 OVERVIEW

The goal of 3D instance segmentation is to determine the categories and binary masks of all foreground objects in the scene. The architecture of our method is illustrated in Figure 3. Assuming that the input point cloud has $N$ points, each point contains position $(x, y, z)$, color $(r, g, b)$ and normal $(n_x, n_y, n_z)$ information. Initially, we utilize a Sparse UNet Contributors (2022) to extract per-point features $F$. Next, we perform farthest point sampling (FPS) on the entire point cloud coordinates to obtain $\mathcal{S}$ sampled points $Q^p$, representing position queries. Subsequently, we input these sampled points $Q^p$ into the Agent-Interpolation Initialization Module (in Section 3.3) to interpolate and obtain corresponding content queries $Q^c$. Finally, we feed $Q^p$ and $Q^c$ together into the Hierarchical Query Fusion Decoder (in Section 3.4) for decoding, resulting in the final instance predictions.

### 3.2 FEATURE EXTRACTION

We employ Sparse UNet as the backbone for feature extraction, yielding features $F \in \mathbb{R}^{N \times C}$, which is consistent with SPFormer Sun et al. (2023) and Maft Lai et al. (2023). Next, we aggregate the point-level features $F$ into superpoint-level features $F_{\text{sup}}$ using average pooling, which will serve as

| X | Y | Z |
|---|---|---|
| 0.2262m | 0.2145m | 0.2367m |

Table 1: **The mean distance between the coordinates of FPS sampling points and the center points of the final predicted instances on ScanNetV2 validation set.**

the key and value for cross-attention in the transformer decoder layer (Section 3.4). Subsequently, we perform FPS on the entire point cloud coordinates to obtain $\mathcal{S}$ sampled points $Q^p$.

### 3.3 AGENT-INTERPOLATION INITIALIZATION MODULE

#### 3.3.1 DISCUSSION

**(a) Position Information:** Our method follows QueryFormer Lu et al. (2023) and Maft Lai et al. (2023), achieving a strong correlation between the positions of sampling points and the positions of the corresponding predicted instances. The details can be found in the supplemental materials A.3. As shown in Table 1, we calculate the mean distance between the coordinates of FPS sampling points and the center points of the final predicted instances. The results show that the distances are small relative to the scale of the scene, validating the strong correlation between the FPS positions and the predicted instance positions. This is why we use FPS to initialize the position embedding of the query—it can sample nearly 100% of foreground instances. In contrast, the learnable-based method of Maft is prone to empty sampling initially. As shown in the second column of Table 5, we have recorded the foreground recall rate of the first layer predictions, which supports the above viewpoint.

**(b) Content Information:** In our method, the primary role of content embedding is to provide a strong global inductive bias. This global inductive bias offers specific information about the dataset: **Firstly**, the dataset being an indoor scene, resulting in biased distributions of point cloud coordinates (XYZ) and color (RGB). **Secondly**, this task is instance segmentation, so the query needs to focus more on positional information (unlike semantic segmentation, which only requires attention to semantics).

And similar to most transformer-based methods, the decoder's input (query) includes position embedding and content embedding. The position embedding represents the query's location in the scene, encoding positional information, while the content embedding is mainly used for subsequent instance prediction by being input into the cls head and mask head for predictions. Notably, in the transformer's attention operation, position information converges into the content embedding. Next, we will introduce several design schemes for the combination of position embedding and content embedding, discussing their advantages and disadvantages.

**FPS + Zero.** This scheme only includes information from a single scene through FPS, lacking the necessary global inductive bias (just like how image preprocessing typically normalizes using the mean and standard deviation of ImageNet Deng et al. (2009)).

**FPS + Learnable.** Although learnable embedding can capture global inductive bias, the positions obtained by FPS for different scenes are entirely different, while the learnable embedding is shared across all scenes. Therefore, there is a lack of correspondence between position and learnable embedding.

**Learnable + Learnable/Zero.** Although this approach ensures correspondence between position embedding and content embedding, it loses the prior knowledge of a single scene. (FPS can obtain the prior of a single scene, i.e., higher foreground coverage for the current scene. Given the wide, sparse, and diverse distribution of point cloud, it is challenging for learnable embedding to cover instances effectively.)

**FPS + Agent (Interpolation)—Our Method.** Firstly, we use FPS to obtain the prior for the current scene. Next, we use interpolation to acquire the global inductive bias. Since the agent contains corresponding position embedding and content embedding, our method balances single scene priors, global inductive bias, and correspondence. To validate this, as shown in the 3 to 5 column of Table 5, we record the APs of the first layer predictions (the main difference among the three setups lies in the content embedding). Our agent-based interpolation method can acquire richer content information (strong global inductive bias), thereby improving the APs metrics.

### 3.3.2 METHOD DETAILS

In this section, we will introduce the process of obtaining content queries through agent interpolation. Firstly, we initialize $L$ agents, which contain $L$ learnable position coordinates $Q_0^p \in [0,1]^{L \times 3}$ and $L$ learnable content queries $Q_0^c \in \mathbb{R}^{L \times C}$. Given the significant variation in the range of points among different scenes, we perform a scene-specific refinement on the normalized $Q_0^p$,

$$\widehat{Q_0^p} = Q_0^p \cdot (p_{max} - p_{min}) + p_{min}, \tag{1}$$

where $p_{max} \in \mathbb{R}^3$, $p_{min} \in \mathbb{R}^3$ represent the maximum and minimum coordinates of the input scene respectively. Next, it's time to interpolate content queries $Q^c$ based on agents and sampled points $Q^p$. Specifically, we first compute the nearest $K$ agents in the $\widehat{Q_0^p}$ set to each sampled point $Q^p$,

$$dis, idx = \text{KNN}(\widehat{Q_0^p}, Q^p), \tag{2}$$

where $dis \in \mathbb{R}^{\mathcal{S} \times K}$, $idx \in \mathbb{N}^{\mathcal{S} \times K}$. Following that, we calculate weights $\mathbf{W} \in [0,1]^{\mathcal{S} \times K}$ based on the distance $dis$,

$$\mathbf{W}_{i,j} = \frac{dis_{i,j}^{-1}}{\sum_{j=1}^{K} dis_{i,j}^{-1}}, \tag{3}$$

where $i, j$ represent the $i$-th sampled point and the $j$-th agent. Finally, we weight $Q_0^c$ to obtain the content queries $Q^c$ corresponding to the sampled points $Q^p$,

$$Q_i^c = \sum_{j=1}^{K} \mathbf{W}_{i,j} \text{Gather}(Q_0^c, idx)_{i,j}, \tag{4}$$

where Gather Paszke et al. (2019) is used to collect values from an input tensor according to specified indices.

After obtaining $Q^c$, we feed $Q^c$ and $Q^p$ together into the Hierarchical Query Fusion Decoder for instance prediction. However, it is worth noting that if we directly feed $Q^p$ in, we cannot update the learnable position coordinates $Q_0^p$ through gradient backpropagation; only $Q_0^c$ can be updated. Therefore, to ensure that $Q_0^p$ can also be continuously updated along with the network training, we make some modifications to $Q^p$,

$$\widehat{Q^p} = \text{SG}(Q^p - \Phi(\mathbf{W}, Q_0^p, idx)) + \Phi(\mathbf{W}, Q_0^p, idx), \tag{5}$$

where SG Van Den Oord et al. (2017) refers to stop gradient, $\Phi$ achieves the same functionality with Equation 4. With this ingenious design, the values of $\widehat{Q^p}$ equal $Q^p$, and $Q_0^p$ remain updatable. To maintain brevity in our writing, we will continue to use $Q^p$ to represent $\widehat{Q^p}$ in subsequent modules.

### 3.4 HIERARCHICAL QUERY FUSION DECODER

The purpose of this section is to generate final instance predictions through decoding. In previous approaches, multiple decoder layers are employed to refine queries. For output queries of each layer, we utilize MLPs to obtain the corresponding instance categories and masks. The acquired instance categories and masks are matched with the ground truth using the Hungarian Matching algorithm Kuhn (1955) and supervised using per-layer auxiliary loss. In this process, the presence of noisy features leads to unstable directions in query optimization, resulting in instability in Hungarian Matching results, especially for those hard-to-predict instances. Consequently, those hard-to-predict instances are difficult to acquire better mask quality through multiple decoder layers, ultimately leading to **Object Disappearance** and decreased recall (as shown in Figure 2 (b)).

Therefore, to mitigate this problem, we merge specific queries from different layers, retaining pre-update queries that exhibit a low overlap compared to post-update queries. Specifically, suppose the queries $Q_{l-1}^p$ and $Q_{l-1}^c$, outputted from the ($l$-1)-th layer, is updated to $Q_l^p$ and $Q_l^c$ after the update in the $l$-th layer. We first calculate the instance masks $\mathbf{M}_{l-1}$ and $\mathbf{M}_l$ corresponding to $Q_{l-1}^c$ and $Q_l^c$. Next, we compute the IoU $\in [0,1]^{\mathcal{S}_{l-1} \times \mathcal{S}_l}$ between $\mathbf{M}_{l-1}$ and $\mathbf{M}_l$. We calculate the maximum IoU between each mask from layer $(l-1)$ and the masks from layer $l$,

$$\mathbf{U}_i = \max_j(\text{IoU}_{i,j}), \tag{6}$$

Finally, we perform a Bottom-K operation on $\mathbf{U}$, selecting the indices $\mathcal{I} \in \mathbb{N}^{\mathcal{D}_1 \times 1}$ corresponding to the smallest $\mathcal{D}_1$ values in $\mathbf{U}$. We utilize the indices $\mathcal{I}$ to retrieve the corresponding queries from the ($l$-1)-th layer. These queries are concatenated with those from the $l$-th layer and collectively fed into the ($l$+1)-th layer. For details regarding the transformer decoder layer, please refer to the supplemental materials.

Through this selection mechanism, queries are given the opportunity for re-updating. If the updated queries perform poorly, the pre-update queries will be retained and passed to the next layer for re-updating. If the update is moderate or reasonably satisfactory, whether to retain the pre-update queries or not is acceptable. Recall also experiences a gradual and steady improvement layer by layer. To be more specific, we introduce the details in the supplemental materials A.3. It is worth noting that the increase in the number of queries imposes a limited burden on runtime, with a 7.8% increase. One final point to add is that since the queries in the earlier layers have not aggregated enough instance information, we do not perform the aforementioned fusion operation. Instead, we only conduct the fusion operation at the final $\mathcal{D}_2$ layers. Here, $\mathcal{D}_2$ indicates the layers where the fusion operation is performed. For example, $\mathcal{D}_2$=3 means we perform the fusion operation in the last 3 layers.

### 3.5 MODEL TRAINING AND INFERENCE

Following Maft Lai et al. (2023), the training loss we utilize contains five aspects,

$$L_{all} = \lambda_1 L_{ce} + \lambda_2 L_{bce} + \lambda_3 L_{dice} + \lambda_4 L_{center} + \lambda_5 L_{score}, \tag{7}$$

where $\lambda_1, \lambda_2, \lambda_3, \lambda_4, \lambda_5$ are hyperparameters. It is worth noting that we apply $L_{all}$ supervision to the output of each layer. During the model inference phase, we use the predictions from the final layer as the final output. In addition to the normal forward pass through the network, we also employ NMS on the final output as a post-processing operation. A further discussion on NMS is provided in the supplementary materials.

## 4 EXPERIMENT

### 4.1 EXPERIMENTAL SETUP

**Dataset and Metrics.** We conduct our experiments on ScanNetV2 Dai et al. (2017), Scan-Net200 Rozenberszki et al. (2022), ScanNet++ Yeshwanth et al. (2023) and S3DIS Armeni et al. (2016) datasets. ScanNetV2 includes 1,613 scenes with 18 instance categories. Among them, 1,201 scenes are used for training, 312 scenes are used for validation, and 100 scenes are used for test. ScanNet200 employs the same point cloud data, but it enhances annotation diversity, covering 200 classes, 198 of which are instance classes. ScanNet++ contains 460 high-resolution (sub-millimeter) indoor scenes with dense instance annotations, including 84 distinct instance categories. S3DIS is a large-scale indoor dataset collected from six different areas. It contains 272 scenes with 13 instance categories. Following previous works Lai et al. (2023), the scenes in Area 5 are used for validation and the others are for training. AP@25 and AP@50 represent the average precision scores with IoU thresholds 25% and 50%, and mAP represents the average of all the APs with IoU thresholds ranging from 50% to 95% with a step size of 5%. On ScanNetV2, we report mAP, AP@50 and AP@25. Moreover, we also report the Box AP@50 and AP@25 results following SoftGroup Vu et al. (2022) and DKNet Wu et al. (2022). On ScanNet200 and ScanNet++, we report mAP, AP@50 and AP@25. On S3DIS, we report AP@50 and AP@25.

**Implementation Details.** On ScanNetV2, we train our model on a single RTX3090 with a batch size of 8 for 512 epochs. We employ Maft Lai et al. (2023) as the baseline architecture, with the backbone and transformer decoder layers identical to Maft's. We employ AdamW Loshchilov & Hutter (2017) as the optimizer and PolyLR as the scheduler, with a maximum learning rate of 0.0002. Point clouds are voxelized with a size of 0.02m. For hyperparameters, we tune $\mathcal{S}, L, K, \mathcal{D}_1, \mathcal{D}_2$ as 400, 400, 3, 40, 3 respectively. $\lambda_1, \lambda_2, \lambda_3, \lambda_4, \lambda_5$ in Equation 7 are set as 0.5, 1, 1, 0.5, 0.5. Additional implementation details for other datasets are presented in the supplemental materials.

### 4.2 COMPARISON WITH EXISTING METHODS.

**Results on ScanNetV2.** Table 2 reports the results on ScanNetV2 validation and hidden test set. Due to our method's design of the Agent-Interpolation Initialization Module, which combines FPS with

| Method | ScanNetV2 validation | | | | | ScanNetV2 test | |
|---|---|---|---|---|---|---|---|
| | mAP | AP@50 | AP@25 | Box AP@50 | Box AP@25 | mAP | AP@50 |
| 3D-SIS Hou et al. (2019) | / | 18.7 | 35.7 | 22.5 | 40.2 | 16.1 | 38.2 |
| 3D-MPA Engelmann et al. (2020) | 35.3 | 51.9 | 72.4 | 49.2 | 64.2 | 35.5 | 61.1 |
| DyCo3D He et al. (2021) | 40.6 | 61.0 | / | 45.3 | 58.9 | 39.5 | 64.1 |
| PointGroup Jiang et al. (2020b) | 34.8 | 56.9 | 71.3 | 48.9 | 61.5 | 40.7 | 63.6 |
| MaskGroup Zhong et al. (2022) | 42.0 | 63.3 | 74.0 | / | / | 43.4 | 66.4 |
| OccuSeg Han et al. (2020) | 44.2 | 60.7 | / | / | / | 48.6 | 67.2 |
| HAIS Chen et al. (2021b) | 43.5 | 64.4 | 75.6 | 53.1 | 64.3 | 45.7 | 69.9 |
| SSTNet Liang et al. (2021) | 49.4 | 64.3 | 74 | 52.7 | 62.5 | 50.6 | 69.8 |
| SoftGroup Vu et al. (2022) | 45.8 | 67.6 | 78.9 | 59.4 | 71.6 | 50.4 | 76.1 |
| DKNet Wu et al. (2022) | 50.8 | 66.9 | 76.9 | 59.0 | 67.4 | 53.2 | 71.8 |
| ISBNet Ngo et al. (2023) | 54.5 | 73.1 | 82.5 | 62.0 | 78.1 | 55.9 | 75.7 |
| Spherical Mask Shin et al. (2024) | **62.3** | **79.9** | **88.2** | / | / | **61.6** | **81.2** |
| Mask3D Schult et al. (2022) | 55.2 | 73.7 | 82.9 | 56.6 | 71.0 | 56.6 | 78.0 |
| QueryFormer Lu et al. (2023) | 56.5 | 74.2 | 83.3 | 61.7 | 73.4 | 58.3 | 78.7 |
| SPFormer Sun et al. (2023) | 56.3 | 73.9 | 82.9 | / | / | 54.9 | 77.0 |
| Maft Lai et al. (2023) | 58.4 | 75.9 | 84.5 | 63.9 | 73.5 | 57.8 | 77.4 |
| Ours | **61.7** | **79.5** | **86.5** | **65.3** | **74.6** | **60.6** | **81.0** |

Table 2: **Comparison on ScanNetV2 validation and hidden test set.** The second and third rows are the non-transformer-based and transformer-based methods, respectively.

| Method | ScanNet++ validation | | | ScanNet++ test | | |
|---|---|---|---|---|---|---|
| | mAP | AP@50 | AP@25 | mAP | AP@50 | AP@25 |
| PointGroup Jiang et al. (2020b) | / | / | / | 8.9 | 14.6 | 21.0 |
| HAIS Chen et al. (2021b) | / | / | / | 12.1 | 19.9 | 29.5 |
| SoftGroup Vu et al. (2022) | / | / | / | 16.7 | 29.7 | 38.9 |
| Maft Lai et al. (2023) | 23.1 | 32.6 | 39.7 | 20.9 | 31.3 | 40.4 |
| Ours | **25.3** | **35.2** | **42.6** | **22.2** | **32.8** | **42.5** |

Table 3: **Comparison on ScanNet++ validation and hidden test set.** ScanNet++ contains denser point cloud scenes and wider instance classes than ScanNetV2, with 84 distinct instance classes.

| Method | mAP | AP@50 | AP@25 |
|---|---|---|---|
| SPFormer Sun et al. (2023) | 25.2 | 33.8 | 39.6 |
| Mask3D Schult et al. (2022) | 27.4 | 37.0 | 42.3 |
| QueryFormer Lu et al. (2023) | 28.1 | 37.1 | 43.4 |
| Maft Lai et al. (2023) | 29.2 | 38.2 | 43.3 |
| Ours | **30.5** | **40.0** | **44.8** |

| Method | Recall@50 | mAP | AP@50 | AP@25 |
|---|---|---|---|---|
| Learnable-based | 82.4 | 39.8 | 51.8 | 58.8 |
| FPS-based | 83.8 | 39.2 | 51.4 | 58.5 |
| Ours | **84.1** | **43.1** | **55.7** | **62.7** |

Table 4: **Comparison on ScanNet200 validation set.** ScanNet200 employs the same point cloud data as ScanNetV2 but enhances more annotation diversity, with 198 instance classes.

Table 5: **Effectiveness of the Agent-Interpolation Initialization Module.** We evaluate the performance of the first layer predictions on ScanNetV2 validation set.

learnable queries to acquire stronger position and content information, as well as the adoption of the Hierarchical Query Fusion Decoder to enhance recall rate, our approach significantly outperforms other transformer-based methods, achieving an increase in mAP by 3.3, AP@50 by 3.6, AP@25 by 2.0, Box AP@50 by 1.4 and Box AP@25 by 1.1 in the validation set, and a rise in mAP by 2.8, AP@50 by 3.6 in the hidden test set. To vividly illustrate the differences between our method and others, we visualize the qualitative results in Figure 4. From the regions highlighted in red boxes, it is evident that our method can generate more accurate predictions.

**Results on ScanNet++.** Table 3 presents the results on ScanNet++ validation and hidden test set. The notable performance enhancement underscores the efficacy of our method in handling denser point cloud scenes.

**Results on ScanNet200.** Table 4 reports the results on ScanNet200 validation set. The significant performance improvement demonstrates the effectiveness of our method in handling such complex scenes with a broader range of categories.

**Results on S3DIS.** We evaluate our method on S3DIS using Area 5 in Table 6. Our proposed method achieves superior performance compared to previous methods, with large margins in both AP@50 and AP@25, demonstrating the effectiveness and generalization of our method.

Figure 4: **Visualization of instance segmentation results on ScanNetV2 validation set.** The red boxes highlight the key regions.

| Method | AP@50 | AP@25 |
|---|---|---|
| PointGroup Jiang et al. (2020b) | 57.8 | / |
| MaskGroup Zhong et al. (2022) | 65.0 | / |
| SoftGroup Vu et al. (2022) | 66.1 | / |
| SSTNet Liang et al. (2021) | 59.3 | / |
| SPFormer Sun et al. (2023) | 66.8 | / |
| Mask3D Schult et al. (2022) | 68.4 | 75.2 |
| QueryFormer Lu et al. (2023) | 69.9 | / |
| Maft Lai et al. (2023) | 69.1 | 75.7 |
| Ours | **71.9** | **77.8** |

Table 6: **Comparison on S3DIS Area5.** S3DIS contains 13 instance categories.

| AI2M | HQFD | NMS | mAP | AP@50 | AP@25 |
|---|---|---|---|---|---|
| ✗ | ✗ | ✗ | 58.4 | 75.2 | 83.5 |
| ✓ | ✗ | ✗ | 60.1 | 78.2 | 85.6 |
| ✗ | ✓ | ✗ | 60.3 | 77.9 | 85.3 |
| ✓ | ✓ | ✗ | 61.1 | 78.2 | 85.6 |
| ✗ | ✗ | ✓ | 59.0 | 76.1 | 84.3 |
| ✓ | ✗ | ✓ | 60.5 | 78.7 | 85.7 |
| ✗ | ✓ | ✓ | 60.9 | 78.1 | 85.7 |
| ✓ | ✓ | ✓ | **61.7** | **79.5** | **86.5** |

Table 7: **Evaluation of the model with different designs on ScanNet-v2 validation set.** AI2M refers to the Agent-Interpolation Initialization Module. HQFD indicates that the Hierarchical Query Fusion Decoder. NMS refers to Non-Maximum Suppression.

| $\mathcal{S}$ | $L$ | $K$ | mAP | AP@50 | AP@25 |
|---|---|---|---|---|---|
| 400 | 400 | 1 | 61.3 | 78.7 | 85.4 |
| 400 | 400 | 3 | **61.7** | **79.5** | 86.5 |
| 400 | 400 | 8 | 61.3 | 79.3 | **86.9** |
| 400 | 800 | 8 | 61.2 | 78.9 | 86.7 |
| 400 | 200 | 3 | 60.7 | 78.0 | 86.1 |
| 200 | 400 | 3 | 59.8 | 77.3 | 85.0 |
| 600 | 400 | 3 | 60.5 | 77.5 | 84.7 |

Table 8: **Ablation study on $\mathcal{S}$, $L$ and $K$ of the Agent-Interpolation Initialization Module.** $\mathcal{S}$ refers to the number of sampled points. $L$ represents the number of agents. $K$ represents the number of neighbours.

## 4.3 ABLATION STUDIES

**Evaluation of the model with different designs.** To further study the effectiveness of our designs, we conduct ablation studies on ScanNet-v2 validation set. As shown in the Table 7, the second row shows that with the help of AI2M, our model acquire a better position and content information, achieving a performance gain of 1.7, 3.0 in mAP and AP@50. The third row demonstrates that with the help of query fusion in HQFD, a performance gain of 1.9, 2.7 has been achieved in mAP and AP@50. The fourth row demonstrates the effective collaboration between AI2M and HQFD, resulting in performance improvement. The last four rows show that with the assistance of NMS, some spurious predictions can be filtered out, leading to enhanced performance.

**Effectiveness of the Agent-Interpolation Initialization Module.** As shown in Table 5, with the assistance of the Agent-Interpolation Initialization Module, there has been an improvement in the foreground coverage of initial queries, subsequently leading to an increase in the recall rate of the first layer predictions, thus enhancing overall performance. Compared to learnable-based methods, it is evident that the recall rate has significantly improved, leading to performance enhancement. Conversely, in comparison to FPS-based methods, although there isn't a substantial difference in the recall rate of the initial layer, the presence of stronger content information contributes to a notable enhancement in performance.

**Ablation study on $\mathcal{S}$, $L$ and $K$ of the Agent-Interpolation Initialization Module.** As depicted in Table 8, it can be inferred that for $\mathcal{S}$, $L$ and $K$, an intermediate value often yields superior results, specifically when set at $\mathcal{S}$=400, $L$=400, and $K$=3. Also, it can be observed that $\mathcal{S}$ and $L$ have a relatively large impact on the results, similar to the conclusions of previous studies Schult et al. (2022); Lai et al. (2023). In contrast, $K$ has a minimal effect on the results, demonstrating the robustness of our method with respect to $K$.

**Effectiveness of the Hierarchical Query Fusion Decoder.** In this section, we conduct multiple experiments to validate the effectiveness and generalization ability of the Hierarchical Query Fusion Decoder (HQFD). Firstly, as shown in the second column of the Table 9, adding HQFD on top of

| Strategy | Num | mAP | AP@50 | AP@25 |
|---|---|---|---|---|
| Baseline | 400 | 58.4 | 75.2 | 83.5 |
| Baseline | 520 | 58.4 | 75.1 | 83.2 |
| Baseline+COE | 400 | 57.3 | 73.5 | 81.8 |
| Baseline+COE | 520 | 57.4 | 74.1 | 81.8 |
| Baseline+HQFD | 520 | **60.3** | **77.9** | **85.3** |

Table 9: **Effectiveness of the Hierarchical Query Fusion Decoder.** Num refers to the number of queries. COE refers to concatenating the outputs of each layer and then conducting NMS.

| Method | mAP | AP@50 | AP@25 |
|---|---|---|---|
| SPFormer[†] Sun et al. (2023) | 57.2 | 75.9 | 83.5 |
| SPFormer[†]+HQFD | 59.4 | 77.8 | 85.5 |
| Maft[†] Lai et al. (2023) | 59.0 | 76.1 | 84.3 |
| Maft[†]+HQFD | **60.9** | **78.1** | **85.7** |

Table 10: **Generalization of the Hierarchical Query Fusion Decoder.** The symbol † indicates the results obtained after adding the NMS operation.

| Method | Parameter(M) | Runtime(ms) |
|---|---|---|
| HAIS Chen et al. (2021b) | 30.9 | 578 |
| SoftGroup Vu et al. (2022) | 30.9 | 588 |
| SSTNet Liang et al. (2021) | / | 729 |
| Mask3D Schult et al. (2022) | 39.6 | 578 |
| QueryFormer Lu et al. (2023) | 42.3 | 487 |
| SPFormer Sun et al. (2023) | 17.6 | 430 |
| Maft Lai et al. (2023) | 20.1 | 412 |
| Ours | 20.3 | 444 |

Table 11: **Parameter and runtime analysis of different methods on ScanNetV2 validation set.** The runtime is measured on the same device.

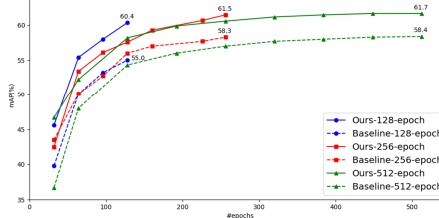

Figure 5: **The convergence curve under different settings on ScanNet-v2 validation set.**

the baseline leads to an increase in the final output queries count. However, this increase is limited and has minimal impact on computational load. Next, we compare the performance of the baseline and the baseline enhanced with HQFD under the same number of queries. The second row of results indicate that simply increasing the number of queries not only does not improve performance but also leads to a slight decrease in performance, which is in contrast to the results of our method in the fifth row. This demonstrates that the performance improvement of our method does not stem from an increase in the number of queries but rather from maintaining a higher recall rate, as can be evidenced in Figure 2 (b). We also report the performance of baseline+COE in the third and fourth rows, and the relevant description is in the third paragraph of Section 1. Results suggest that simply adopting the COE operation does not enhance performance, but leads to a decline. Our method of progressively retaining queries with low overlap can significantly improve performance.

To demonstrate the generalization capability of HQFD, we also add HQFD to other methods, as shown in Table 10. The performance improvement on SPFormer and Maft effectively demonstrates that our method can serve as a plug-and-play module for other transformer-based methods.

**Contribution to the convergence speed.** As shown in Figure 5, with only 128-epoch training, our method outperforms the baseline trained with 512 epochs. This can be attributed to AI2M ensuring high foreground coverage of initial queries, along with HQFD ensuring a steady increase in recall during the decoding process.

### 4.4 PARAMETER AND RUNTIME ANALYSIS.

Table 11 reports the model parameter and the runtime per scan of different methods on ScanNetV2 validation set. For a fair comparison, the reported runtime is measured on the same RTX 3090 GPU. Compared with Maft, our method achieves noticeable performance improvement with a 0.2M parameter increment. As to the inference speed, our method is faster than most methods. Performance, parameter efficiency, and speed collectively demonstrate our method's efficacy, practicality, and applicability.

## 5 CONCLUSION

In this paper, we propose a novel 3D instance segmentation method termed BFL. To generate queries capable of achieving a nuanced balance between foreground coverage and content learning, we promose the Agent-Interpolation Initialization Module. Furthermore, the well-designed Hierarchical Query Fusion Decoder mitigates the decrease in recall with the deepening of layers. Extensive experiments conducted on the several datasets demonstrate the superior performance of BFL.

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

## A   APPENDIX

You may include other additional sections here.

### A.1   OVERVIEW

This supplementary material provides more model and experimental details to understand our proposed method. After that, we present more experiments to demonstrate the effectiveness of our methods. Finally, we show a rich visualization of our modules.

### A.2   MORE MODEL DETAILS

**Sparse UNet.** For ScanNetV2 Dai et al. (2017), ScanNet200 Rozenberszki et al. (2022), and ScanNet++ Yeshwanth et al. (2023), we employ a 5-layer U-Net as the backbone, with the initial channel set to 32. Unless otherwise specified, we utilize coordinates, colors, and normals as input features. Our method incorporates 6 layers of Transformer decoders, with the head number set to 8, and the hidden and feed-forward dimensions set to 256 and 1024, respectively. For S3DIS Armeni et al. (2016), following Mask3D Schult et al. (2022), we utilize Res16UNet34C Choy et al. (2019) as the backbone and employ 4 decoders to attend to the coarsest four scales. This process is repeated 3 times with shared parameters. The dimensions for the decoder's hidden layer and feed-forward layer are set to 128 and 1024, respectively.

**Transformer Decoder Layer.** In this layer, we use superpoint-level features $F_{\text{sup}}$ and their corresponding positions $P_{\text{sup}}$ as key and value, with content queries $Q^c$ and position queries $Q^p$ as query. The specific network architecture can be seen in Figure 6, which is identical to Maft's Lai et al. (2023) transformer decoder layer. Therefore, more relevant equations and details can be directly referred to Maft's main text.

Figure 6: **The architecture of the transformer decoder layer.** The figure is taken from the main text of Maft.

**Matching and Loss.** Existing methods depend on semantic predictions and binary masks for matching queries with ground truths. Building upon Maft Lai et al. (2023), our approach integrates center distance into Hungarian Matching Kuhn (1955). To achieve this, we modify the formulation of matching costs as follows:

$$\mathcal{C}_{cls}(p, \overline{p}) = CE(CLASS_p, CLASS_{\overline{p}}), \tag{8}$$

$$\mathcal{C}_{dice}(p, \overline{p}) = DICE(MASK_p, MASK_{\overline{p}}), \tag{9}$$

$$\mathcal{C}_{bce}(p, \overline{p}) = BCE(MASK_p, MASK_{\overline{p}}), \tag{10}$$

$$\mathcal{C}_{center}(p, \overline{p}) = L1(Center_p, Center_{\overline{p}}), \tag{11}$$

$$\mathcal{C}(p, \overline{p}) = \lambda_{cls}\mathcal{C}_{cls}(p, \overline{p}) + \lambda_{dice}\mathcal{C}_{dice}(p, \overline{p}) + \lambda_{bce}\mathcal{C}_{bce}(p, \overline{p}) + \lambda_{center}\mathcal{C}_{center}(p, \overline{p}), \tag{12}$$

where $p$ and $\overline{p}$ denotes a predicted and ground-truth instance, $\mathcal{C}$ represents the matching cost matrix, and $\lambda_{cls}, \lambda_{dice}, \lambda_{bce}, \lambda_{center}$ are the hyperparameters. Here, $\lambda_{cls}, \lambda_{dice}, \lambda_{bce}, \lambda_{center}$ are the same as $\lambda_1, \lambda_2, \lambda_3, \lambda_4$. Next, we perform Hungarian Matching on $\mathcal{C}$, and then supervise the Hungarian Matching results according to Equation 7

**Non-Maximum Suppression.** Non-maximum suppression (NMS) is a common post-processing operation used in instance segmentation. In fact, for some previous methods, applying NMS to the final layer predictions has consistently led to performance improvements, as shown in Table 12. However, if we apply NMS to the concatenated outputs, as described in Section 1 lines 63-65, a significant decrease in performance occur. The specific reasons for this performance decrease are twofold. Firstly, NMS heavily relies on confidence scores, retaining only the masks with the highest confidence among the duplicates. However, these confidence scores are often inaccurate, leading to the retention of masks that are not necessarily of the best quality. Since the concatenated outputs contain a large number of duplicate masks (almost every mask has duplicates), this results in a significant reduction in performance. Secondly, NMS requires manual selection of a threshold. If the threshold is set too high, it cannot effectively filter out duplicate masks; if it is set too low, it tends to discard useful masks. The more complex the output, the more challenging it becomes to select an optimal threshold. Therefore, for concatenated outputs, it is difficult to find an optimal threshold for effective filtering.

| Method | mAP | AP@50 | AP@25 |
|---|---|---|---|
| SPFormer | 56.7 | 74.8 | 82.9 |
| SPFormer+NMS | 57.2 | 75.9 | 83.5 |
| SPFormer+COE | 55.7 | 73.4 | 81.8 |
| Maft | 58.4 | 75.2 | 83.5 |
| Maft+NMS | 59.0 | 76.1 | 84.3 |
| SPFormer+COE | 57.3 | 73.5 | 81.8 |
| Ours | 61.1 | 78.2 | 85.6 |
| Ours+NMS | **61.7** | **79.5** | **86.5** |

Table 12: **The effectiveness of the NMS.** COE refers to concatenating the outputs of each layer and then conducting NMS.

A.3 MORE DISCUSSION

**Details on achieving a strong correlation.** The positions of sampling points in Mask3D are not related to the positions of the corresponding predicted instances. In fact, this lack of correlation results in the query's lack of interpretability, we cannot clearly understand why this query predicts this object, thus hindering intuitive optimization. Both QueryFormer and Maft address this by adding a $\mathcal{C}_{center}$ term when calculating the Hungarian matching cost matrix, which represents the distance between the query coordinates and the ground truth instance center. Additionally, they update the query coordinates layer by layer, making the matched query progressively closer to the GT instance center. With this design, the position of the query becomes correlated with the position of the corresponding predicted instance, facilitating intuitive improvements in the distribution of query initialization by QueryFormer and Maft (Query Refinement Module and Learnable Position Query).

**Detail classification on Hierarchical Query Fusion Decoder.** We aim to give poorly updated queries a new opportunity for updating. It is important to note that this is a copy operation, so we retain both pre-updated and post-updated queries, thus not "limiting the transformer decoder in its ability to swap objects." This approach provides certain queries with an opportunity for entirely new feature updates and offers more diverse matching options during Hungarian matching. This re-updating and diverse selection mechanism clearly enhances recall rates because our design implicitly includes a mechanism: for instances that are difficult to predict or poorly predicted, if the updates are particularly inadequate, the corresponding queries will be retained and accumulated into the final predictions. For example, if a query $Q_i^3$ from the third layer is updated in the fourth layer to become $Q_i^4$ and experiences a significant deviation, the network will retain $Q_i^3$ and pass both $Q_i^3$ and $Q_i^4$ to the fifth layer. After being updated in the fifth layer, $Q_i^3$ becomes $\hat{Q}_i^3$. If $\hat{Q}_i^3$ does not significantly differ from $Q_i^3$, the model will not retain $Q_i^3$ further and will only pass $\hat{Q}_i^3$ to the sixth layer. If $\hat{Q}_i^3$ shows a significant difference from $Q_i^3$, the model will continue to retain $Q_i^3$. Through this process, teh model can continuously retain the queries that are poorly updated, accumulating them into the final prediction.

A.4 MORE IMPLEMENTATION DETAILS

On ScanNet200 Rozenberszki et al. (2022), we train our model on a single RTX3090 with a batch size of 8 for 512 epochs. We employ AdamW Loshchilov & Hutter (2017) as the optimizer and PolyLR as the scheduler, with a maximum learning rate of 0.0002. Point clouds are voxelized with a size of 0.02m. For hyperparameters, we tune $\mathcal{S}, L, K, \mathcal{D}_1, \mathcal{D}_2$ as 500, 500, 3, 40, 3 respectively. $\lambda_1, \lambda_2, \lambda_3, \lambda_4, \lambda_5$ in Equation 7 are set as 0.5, 1, 1, 0.5, 0.5. On ScanNet++ Yeshwanth et al. (2023), we train our model on a single RTX3090 with a batch size of 4 for 512 epochs. The other settings are the same as ScanNet200. On S3DIS Armeni et al. (2016), we train our model on a single A6000 with a batch size of 4 for 512 epochs and adopt onecycle scheduler. For hyperparameters, we tune $\mathcal{S}, L, K, \mathcal{D}_1, \mathcal{D}_2$ as 400, 400, 3, 40, 3 respectively. $\lambda_1, \lambda_2, \lambda_3, \lambda_4, \lambda_5$ in Equation 7 are set as 2, 5, 1, 0.5, 0.5.

A.5 DETAILED RESULTS

The detailed results for each category on ScanNetV2 validation set are reported in Table 13. As the table illustrates, our method achieves the best performance in 16 out of 18 categories. The detailed results for certain categories on ScanNet++ test set are presented in Table 17. As indicated by the table, the significant performance improvement highlights the effectiveness of our method in managing denser point cloud scenes across a broader range of categories.

A.6 MORE ABLATION STUDIES

**Difference in Recall and AP across different decoder layers.** As depicted in Table 18, we conduct an ablation study on ScanNetV2 validation set to examine the impact of our proposed HQFD on recall and AP. From the table, it is evident that the recall of Maft decreases at the fifth layer, consequently leading to a decline in the corresponding AP and influencing the final prediction results. In contrast, our approach, which incorporates HQFD, ensures a steady improvement in recall, thereby

| Method | mAP | bathtub | bed | bookshe. | cabinet | chair | counter | curtain | desk | door | other | picture | frige | s. curtain | sink | sofa | table | toilet | window |
|---|---|---|---|---|---|---|---|---|---|---|---|---|---|---|---|---|---|---|---|
| SoftGroup Vu et al. (2022) | 45.8 | 66.6 | 48.4 | 32.4 | 37.7 | 72.3 | 14.3 | 37.6 | 27.6 | 35.2 | 42.0 | 34.2 | 56.2 | 56.9 | 39.6 | 47.6 | 54.1 | 88.5 | 33.0 |
| DKNet Wu et al. (2022) | 50.8 | 73.7 | 53.7 | 36.2 | 42.6 | 80.7 | 22.7 | 35.7 | 35.1 | 42.7 | 46.7 | 51.9 | 39.9 | 57.2 | 52.7 | 52.4 | 54.2 | 91.3 | 37.2 |
| Mask3D Schult et al. (2022) | 55.2 | 78.3 | 54.3 | 43.5 | 47.1 | 82.9 | 35.9 | 48.7 | 37.0 | 54.3 | 59.7 | 53.3 | 47.7 | 47.4 | 55.6 | 48.7 | 63.8 | 94.6 | 39.9 |
| QueryFormer Lu et al. (2023) | 56.5 | 81.3 | 57.7 | 45.0 | 47.2 | 82.0 | 37.2 | 43.2 | 43.3 | 54.5 | 60.5 | 52.6 | 54.1 | 62.7 | 52.4 | 49.9 | 60.5 | 94.7 | 37.4 |
| Maft Lai et al. (2023) | 58.4 | 80.1 | 58.1 | 41.8 | 48.3 | 82.2 | 34.4 | 55.1 | **44.3** | 55.0 | 57.9 | 61.6 | 56.4 | 63.7 | 54.4 | 53.0 | 66.3 | **95.3** | **42.9** |
| Ours | **61.7** | **83.5** | **62.3** | **48.1** | **50.6** | **84.1** | **45.0** | **57.4** | 42.1 | **57.3** | **61.8** | **67.8** | **59.9** | **68.8** | **61.1** | **55.3** | **66.6** | **95.3** | 42.6 |

Table 13: **Full quantitative results of mAP on ScanNetV2 validation set.** Best performance is in boldface.

| Method | mAP | bathtub | bed | bookshe. | cabinet | chair | counter | curtain | desk | door | other | picture | frige | s. curtain | sink | sofa | table | toilet | window |
|---|---|---|---|---|---|---|---|---|---|---|---|---|---|---|---|---|---|---|---|
| PointGroup Jiang et al. (2020b) | 40.7 | 63.9 | 49.6 | 41.5 | 24.3 | 64.5 | 2.1 | 57.0 | 11.4 | 21.1 | 35.9 | 21.7 | 42.8 | 66.6 | 25.6 | 56.2 | 34.1 | 86.0 | 29.1 |
| MaskGroup Zhong et al. (2022) | 43.4 | 77.8 | 51.6 | 47.1 | 33.0 | 65.8 | 2.9 | 52.6 | 24.9 | 25.6 | 40.0 | 30.9 | 38.4 | 29.6 | 36.8 | 57.5 | 42.5 | 87.7 | 36.2 |
| OccuSeg Han et al. (2020) | 48.6 | 80.2 | 53.6 | 42.8 | 36.9 | 70.2 | 20.5 | 33.1 | 30.1 | 37.9 | 47.4 | 32.7 | 43.7 | **86.2** | 48.5 | 60.1 | 39.4 | 84.6 | 27.3 |
| HAIS Chen et al. (2021b) | 45.7 | 70.4 | 56.1 | 45.7 | 36.4 | 67.3 | 4.6 | 54.7 | 19.4 | 30.8 | 42.6 | 28.8 | 45.4 | 71.1 | 26.2 | 56.3 | 43.4 | 88.9 | 34.4 |
| SSTNet Liang et al. (2021) | 50.6 | 73.8 | 54.9 | 49.7 | 31.6 | 69.3 | 17.8 | 37.7 | 19.8 | 33.0 | 46.3 | 57.6 | 51.5 | 85.7 | **49.4** | 63.7 | 45.7 | 94.3 | 29.0 |
| DKNet Wu et al. (2022) | 53.2 | 81.5 | 62.4 | **51.7** | 37.7 | 74.9 | 10.7 | 50.9 | 30.4 | 43.7 | 47.5 | 58.1 | 53.9 | 77.5 | 33.9 | 64.0 | 50.6 | 90.1 | 38.5 |
| SPFormer Sun et al. (2023) | 54.9 | 74.5 | 64.0 | 48.4 | 39.5 | 73.9 | **31.1** | 56.6 | 33.5 | 46.8 | 49.2 | 55.5 | 47.8 | 74.7 | 43.6 | 71.2 | 54.0 | 89.3 | 34.3 |
| Maft Lai et al. (2023) | 59.6 | 88.9 | **72.1** | 44.8 | 46.0 | **76.8** | 25.1 | 55.8 | **40.8** | 50.4 | 53.9 | 61.6 | 61.8 | 85.8 | 48.2 | 68.4 | 55.1 | 93.1 | **45.0** |
| Ours | 60.6 | **92.6** | 70.2 | 51.5 | **50.2** | 73.2 | 28.2 | **59.8** | 38.6 | 48.9 | **54.2** | 63.5 | **71.6** | 75.1 | 47.6 | **74.3** | **58.7** | **95.8** | 36.0 |

Table 14: **Full quantitative results of mAP on the ScanNetV2 test set. Best performance is in boldface.**

| Method | AP@50 | bathtub | bed | bookshe. | cabinet | chair | counter | curtain | desk | door | other | picture | frige | s. curtain | sink | sofa | table | toilet | window |
|---|---|---|---|---|---|---|---|---|---|---|---|---|---|---|---|---|---|---|---|
| PointGroup Jiang et al. (2020b) | 63.6 | 100.0 | 76.5 | 62.4 | 50.5 | 79.7 | 11.6 | 69.6 | 38.4 | 44.1 | 55.9 | 47.6 | 59.6 | 100.0 | 66.6 | 75.6 | 55.6 | 99.7 | 51.3 |
| MaskGroup Zhong et al. (2022) | 66.4 | 100.0 | 82.2 | 76.4 | 61.6 | 81.5 | 13.9 | 69.4 | 59.7 | 45.9 | 56.6 | 59.9 | 60.0 | 51.6 | 71.5 | 81.9 | 63.5 | 100.0 | 60.3 |
| OccuSeg Han et al. (2020) | 67.2 | 100.0 | 75.8 | 68.2 | 57.6 | 84.2 | 47.7 | 50.4 | 52.4 | 56.7 | 58.5 | 45.1 | 55.7 | 100.0 | 75.1 | 79.7 | 56.3 | 100.0 | 46.7 |
| HAIS Chen et al. (2021b) | 69.9 | 100.0 | 84.9 | 82.0 | 67.5 | 80.8 | 27.9 | 75.7 | 46.5 | 51.7 | 59.6 | 55.9 | 60.0 | 100.0 | 65.4 | 76.7 | 67.6 | 99.4 | 56.0 |
| SSTNet Liang et al. (2021) | 69.8 | 100.0 | 69.7 | **88.8** | 55.6 | 80.3 | 38.7 | 62.6 | 41.7 | 55.6 | 58.5 | 70.2 | 60.0 | 100.0 | **82.4** | 72.0 | 69.2 | 100.0 | 50.9 |
| DKNet Wu et al. (2022) | 71.8 | 100.0 | 81.4 | 78.2 | 61.9 | 87.2 | 22.4 | 75.1 | 56.9 | 67.7 | 58.5 | 72.4 | 63.3 | 98.1 | 51.9 | 81.9 | 73.6 | 100.0 | 61.7 |
| SPFormer Sun et al. (2023) | 77.0 | 90.3 | 90.3 | 80.6 | 60.9 | 88.6 | 56.8 | **81.5** | 70.5 | 71.1 | 65.5 | 65.2 | 68.5 | 100.0 | 78.9 | 80.9 | 77.6 | 100.0 | 58.3 |
| Maft Lai et al. (2023) | 78.6 | 100.0 | 89.4 | 80.7 | 69.4 | **89.3** | 48.6 | 67.4 | 74.0 | **78.6** | 70.4 | **72.7** | 73.9 | 100.0 | 70.7 | 84.9 | 75.6 | 100.0 | **68.5** |
| Ours | **81.0** | **100.0** | **93.4** | 85.4 | **74.3** | 88.9 | **57.5** | 71.4 | **81.0** | 66.9 | **72.9** | 70.7 | **80.9** | 100.0 | 81.4 | **90.2** | **81.4** | 100.0 | 62.5 |

Table 15: **Full quantitative results of AP@50 on the ScanNetV2 test set. Best performance is in boldface.**

| Method | AP@25 | bathtub | bed | bookshe. | cabinet | chair | counter | curtain | desk | door | other | picture | frige | s. curtain | sink | sofa | table | toilet | window |
|---|---|---|---|---|---|---|---|---|---|---|---|---|---|---|---|---|---|---|---|
| PointGroup Jiang et al. (2020b) | 77.8 | 100.0 | 90.0 | 79.8 | 71.5 | 86.3 | 49.3 | 70.6 | 89.5 | 56.9 | 70.1 | 57.6 | 63.9 | 100.0 | 88.0 | 85.1 | 71.9 | 99.7 | 70.9 |
| MaskGroup Zhong et al. (2022) | 79.2 | 100.0 | 96.8 | 81.2 | 76.6 | 86.4 | 46.0 | 81.5 | 88.8 | 59.8 | 65.1 | 63.9 | 60.0 | 91.8 | 94.1 | 89.6 | 72.1 | 100.0 | 72.3 |
| OccuSeg Han et al. (2020) | 74.2 | 100.0 | 92.3 | 78.5 | 74.5 | 86.7 | 55.7 | 57.8 | 72.9 | 67.0 | 64.4 | 48.8 | 55.7 | 100.0 | 79.4 | 83.0 | 62.0 | 100.0 | 55.0 |
| HAIS Chen et al. (2021b) | 80.3 | 100.0 | **99.4** | 82.0 | 75.9 | 85.5 | 55.4 | 88.2 | 82.7 | 61.5 | 67.6 | 63.8 | 64.6 | 100.0 | 91.2 | 79.7 | 76.7 | 99.4 | 72.6 |
| SSTNet Liang et al. (2021) | 78.9 | 100.0 | 84.0 | **88.8** | 71.7 | 83.5 | 71.7 | 68.4 | 62.7 | 72.4 | 65.2 | 72.7 | 60.0 | 100.0 | 91.2 | 82.2 | 75.7 | 100.0 | 69.1 |
| DKNet Wu et al. (2022) | 81.5 | 100.0 | 93.0 | 84.4 | 76.5 | 91.5 | 53.4 | 80.5 | 80.5 | 80.7 | 65.4 | 76.3 | 65.0 | 100.0 | 79.4 | 88.1 | 76.6 | 100.0 | 75.8 |
| SPFormer Sun et al. (2023) | 85.1 | 100.0 | 99.4 | 80.6 | 77.4 | 94.2 | 63.7 | **84.9** | 85.9 | 88.9 | 72.0 | 73.0 | 66.5 | 100.0 | 91.1 | 86.8 | 87.3 | 100.0 | 79.6 |
| Maft Lai et al. (2023) | 86.0 | 100.0 | 99.0 | 81.0 | 82.9 | **94.9** | **80.9** | 68.8 | 83.6 | **90.4** | 75.1 | **79.6** | 74.1 | 100.0 | 86.4 | 84.8 | 83.7 | 100.0 | **82.8** |
| Ours | **88.2** | **100.0** | 97.9 | 88.2 | **87.9** | 93.7 | 70.3 | 74.9 | **91.5** | 87.5 | **79.5** | 74.0 | **82.0** | 100.0 | **99.4** | **92.3** | **89.1** | 100.0 | 78.8 |

Table 16: **Full quantitative results of AP@25 on the ScanNetV2 test set. Best performance is in boldface.**

guaranteeing a consistent enhancement in AP. This favorable effect on the final output results is attributed to the design of this moudle.

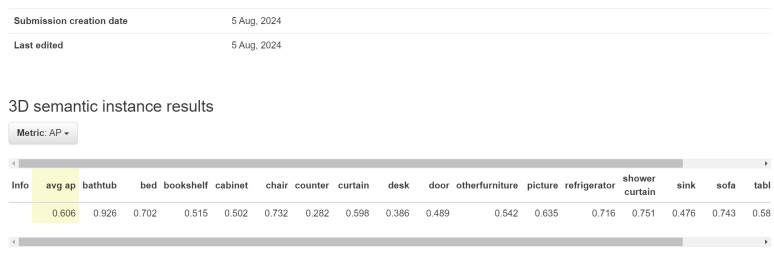

Figure 7: **The mAP result of our method on ScanNetV2 test set.**

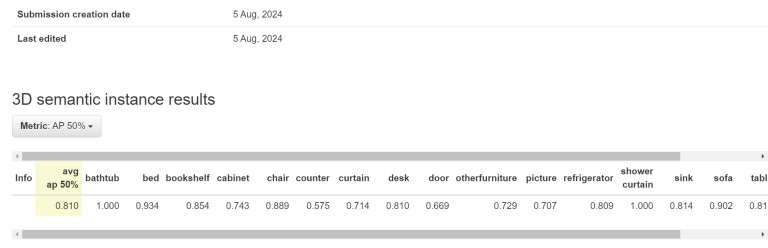

Figure 8: **The AP@50 result of our method on ScanNetV2 test set.**

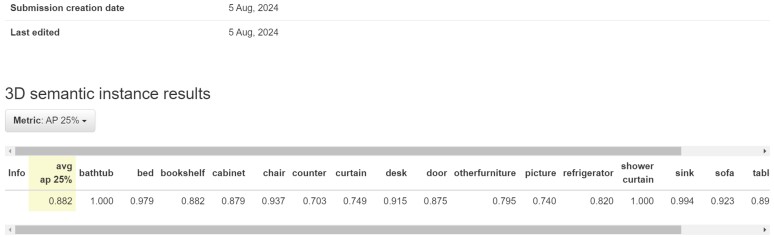

Figure 9: **The AP@25 result of our method on ScanNetV2 test set.**

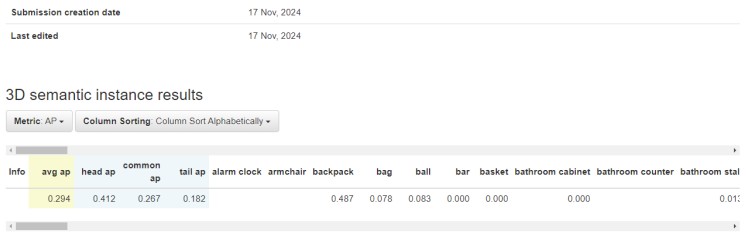

Figure 10: **The mAP result of our method on ScanNet200 test set.**

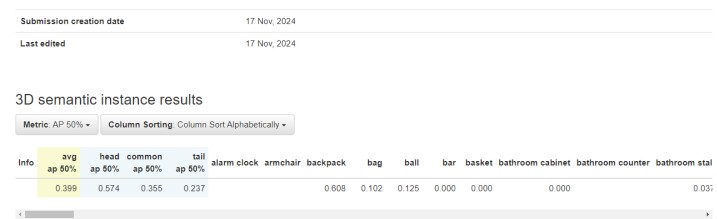

Figure 11: **The AP@50 result of our method on ScanNet200 test set.**

| Submission creation date | 17 Nov, 2024 |
| Last edited | 17 Nov, 2024 |

3D semantic instance results

Metric: AP 25% ▾    Column Sorting: Column Sort Alphabetically ▾

| Info | avg ap 25% | head ap 25% | common ap 25% | tail ap 25% | alarm clock | armchair | backpack | bag | ball | bar | basket | bathroom cabinet | bathroom counter | bathroom stal |
|---|---|---|---|---|---|---|---|---|---|---|---|---|---|---|
| | 0.446 | 0.656 | 0.385 | 0.262 | | | 0.642 | 0.107 | 0.125 | 0.000 | 0.000 | | 0.000 | 0.62 |

Figure 12: **The AP@25 result of our method on ScanNet200 test set.**

| Method | mAP | bottle | box | ceiling l. | cup | monitor | office c. | white. e. | tv | white. | telephone | tap | tissue b. | trash c. | window | sofa | pillow | plant | ... |
|---|---|---|---|---|---|---|---|---|---|---|---|---|---|---|---|---|---|---|---|
| PointGroup Wu et al. (2022) | 8.9 | 0.8 | 2.1 | 57.3 | 13.2 | 37.8 | 82.8 | 0 | 39.0 | 54.7 | 0 | 0 | 0 | 37.2 | 3.5 | 35.7 | 10.1 | 22.5 | ... |
| HAIS Schult et al. (2022) | 12.1 | 3.4 | 3.8 | 55.9 | 16.8 | 49.5 | **87.1** | 0 | 64.1 | 72.5 | 7.2 | 0 | 0 | 29.5 | 4.0 | 49.0 | 14.9 | 25.0 | ... |
| SoftGroup Vu et al. (2022) | 16.7 | 9.4 | 6.2 | 46.7 | 23.2 | 42.8 | 81.3 | 0 | 67.3 | 71.6 | 10.9 | 14.0 | **2.9** | 32.9 | 8.1 | 46.4 | 17.0 | **60.0** | ... |
| Ours | **22.2** | **13.2** | **12.7** | **63.7** | **38.1** | **69.3** | 86.0 | **38.9** | **90.6** | **86.8** | **26.7** | **20.6** | 2.0 | **60.0** | **9.4** | **63.7** | **45.3** | 52.5 | ... |

Table 17: **Full quantitative results of mAP on ScanNet++ test set.** Best performance is in boldface.

| Layer | Ours | | | | Maft | | | |
|---|---|---|---|---|---|---|---|---|
| | Recall@50 | mAP | AP@50 | AP@25 | Recall@50 | mAP | AP@50 | AP@25 |
| 3 | 87.5 | 59.4 | 76.7 | 84.9 | 85.7 | 56.9 | 73.9 | 82.5 |
| 4 | 87.8 (+) | 59.7 (+) | 77.1 (+) | 85.1 (+) | 86.6 (+) | 58.5 (+) | 75.5 (+) | 83.7 (+) |
| 5 | 87.9 (+) | 59.9 (+) | 77.3 (+) | 85.3 (+) | 85.8 (-) | 58.2 (-) | 75.0 (-) | 83.5 (-) |
| 6 | **88.1** (+) | **60.9** (+) | **78.1** (+) | **85.7** (+) | 86.6 (+) | 59.0 (+) | 76.1 (+) | 84.3 (+) |

Table 18: **Difference in Recall and AP across different decoder layers.** (+) indicates an increase compared to the previous layer, while (-) indicates a decrease compared to the previous layer.

**Ablation study on $\mathcal{D}_1$ and $\mathcal{D}_2$ of the Hierarchical Query Fusion Decoder.** $D_1$ represents the number of new added queries in each layer compared to the previous layer, while $D_2$ indicates the layers where the fusion operation is performed. From the table data, we can see that performance decreases significantly when $D_2$=4 compared to $D_2$=3. As analyzed in lines 334-336 in the main text, the queries in the earlier layers have not aggregated enough instance information. Therefore, if $D_2$=4, it means that the queries in the second layer will also participate in the fusion operation, but these queries have only undergone two rounds of feature aggregation, resulting in inaccurate mask predictions. This can affect the operation of the Hierarchical Query Fusion Decoder (HQFD). To ensure the effectiveness of HQFD, we recommend performing the fusion operation on the last half of the decoder layers. In fact, we follow this approach in other datasets as well.

| $\mathcal{D}_1$ | $\mathcal{D}_2$ | mAP | AP@50 | AP@25 |
|---|---|---|---|---|
| 50 | 2 | 61.4 | 78.9 | 86.1 |
| 50 | 3 | 61.5 | 79.2 | 86.3 |
| 50 | 4 | 61.0 | 78.5 | 85.6 |
| 40 | 3 | **61.7** | **79.5** | **86.5** |
| 60 | 3 | 61.3 | 78.8 | 85.9 |

Table 19: **Ablation study on $\mathcal{D}_1$ and $\mathcal{D}_2$ of the Hierarchical Query Fusion Decoder.**

**The effectiveness of the SG in Equation 5.** As illustrated in Table 20, we performed an ablation study on ScanNetV2 validation set to examine the impact of the SG operation in Equation 5. If we do not utilize SG, $Q_0^p$ remains fixed, which hinders its ability to adaptively learn a distribution suitable for all scenarios, thus impacting the overall performance.

| Setting | mAP | AP@50 |
|---|---|---|
| W SG | 61.4 | 79.0 |
| W/o SG | **61.7** | **79.5** |

Table 20: **The effectiveness of the SG in Equation 5.**

**Ablation Study on the hyperparameters in Equation 7.** We perform the experiment in Table 21. Based on the results, we find that the combination 0.5, 1, 1, 0.5, 0.5 yields the best performance.

| $\lambda_1$ | $\lambda_2$ | $\lambda_3$ | $\lambda_4$ | $\lambda_5$ | mAP |
|---|---|---|---|---|---|
| 1 | 1 | 1 | 0.5 | 0.5 | 61.1 |
| 0.5 | 1 | 1 | 0.5 | 0.5 | **61.7** |
| 1.5 | 1 | 1 | 0.5 | 0.5 | 61.4 |
| 0.5 | 0.5 | 1 | 0.5 | 0.5 | 60.8 |
| 0.5 | 1.5 | 1 | 0.5 | 0.5 | 61.5 |
| 0.5 | 1 | 0.5 | 0.5 | 0.5 | 61.0 |
| 0.5 | 1 | 1.5 | 0.5 | 0.5 | 61.2 |
| 0.5 | 1 | 1 | 1 | 0.5 | 61.0 |
| 0.5 | 1 | 1 | 0.5 | 1 | 61.5 |

Table 21: **Ablation Study on the hyperparameters in Equation 7 on ScanNetV2 validation set.**

## A.7 ASSETS AVAILABILITY

The datasets that support the findings of this study are available in the following repositories:

ScanNetV2 Dai et al. (2017) at `http://www.scan-net.org/changelog#scannet-v2-2018-06-11` under the ScanNet Terms of Use. ScanNet200 Rozenberszki et al. (2022) at `https://github.com/ScanNet/ScanNet` under the ScanNet Terms of Use. ScanNet++ Yeshwanth et al. (2023) at `https://kaldir.vc.in.tum.de/scannetpp` under the ScanNet++ Terms of Use. S3DIS Armeni et al. (2016) at `http://buildingparser.stanford.edu/dataset.html` under Apache-2.0 license. The code of our baseline Lai et al. (2023); Sun et al. (2023) is available at `https://github.com/dvlab-research/Mask-Attention-Free-Transformer` and `https://github.com/sunjiahao1999/SPFormer` under MIT license.

## A.8 MORE VISUAL COMPARISON

In Figure 13, we visualize and compare the results of several methods. As shown in this figure's red boxes, our method produces finer segmentation results.

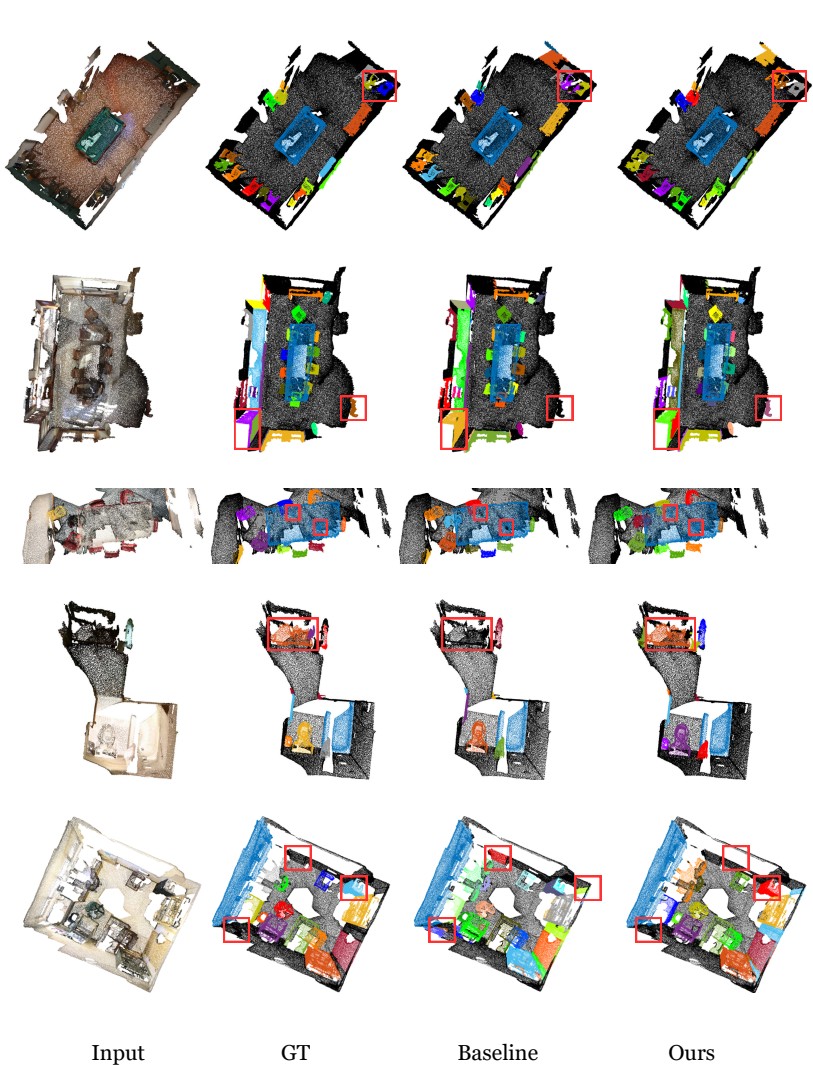

Input       GT       Baseline       Ours

Figure 13: **Additional Visual Comparison on ScanNetV2 validation set.** The red boxes highlight the key regions.

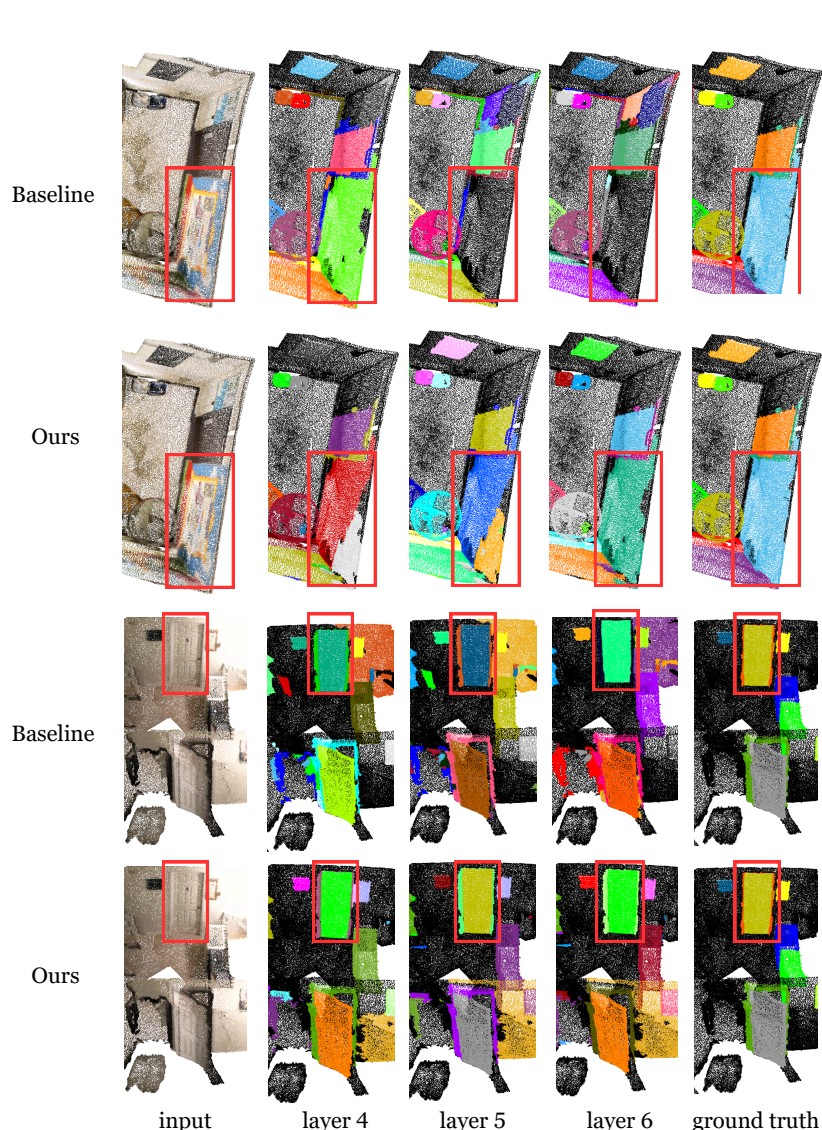

input  layer 4  layer 5  layer 6  ground truth

Figure 14: **Visual comparisons between the baseline and our method across different decoder layers on ScanNetV2 validation set.** The red boxes highlight the key regions.

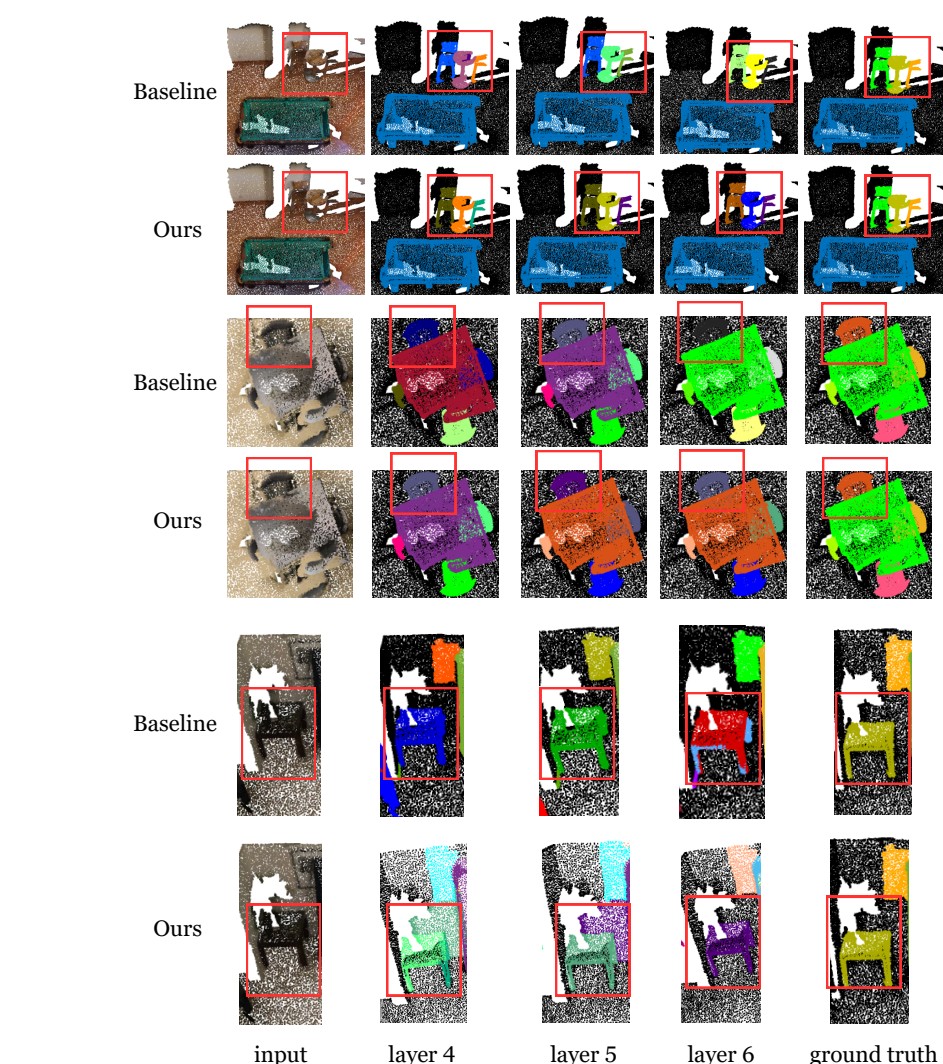

input    layer 4    layer 5    layer 6    ground truth

Figure 15: **Visual comparisons between the baseline and our method across different decoder layers on ScanNetV2 validation set.** The red boxes highlight the key regions.

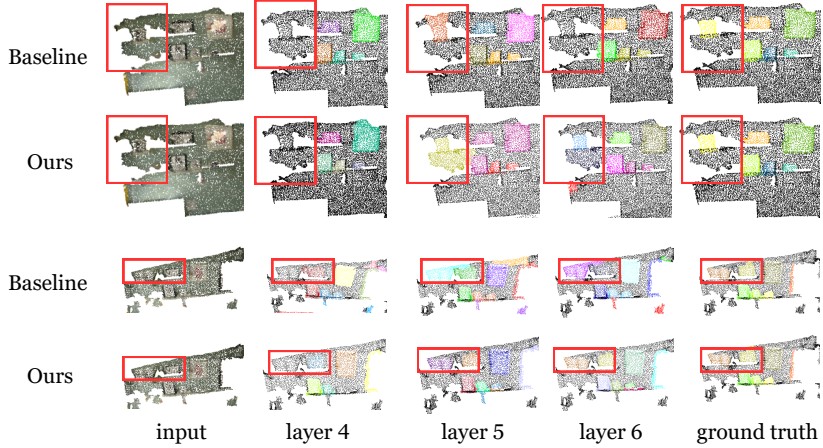

input    layer 4    layer 5    layer 6    ground truth

Figure 16: **Visual comparisons between the baseline and our method across different decoder layers on ScanNetV2 validation set.** The red boxes highlight the key regions.

