# OpenReview forum: "Beyond the Final Layer: Hierarchical Query Fusion Transformer with Agent-Interpolation Initialization for 3D Instance Segmentation"
_ICLR.cc/2025/Conference — Submitted to ICLR 2025_

### Official Review · Reviewer_VSMc · 2024-10-30

**Soundness:** 2
**Presentation:** 2
**Contribution:** 2
**Rating:** 3
**Confidence:** 4

**Summary:**

The authors introduce a 3D instance segmentation framework, Beyond the Final Layer (BFL), to overcome the challenges of existing transformer-based methods.

BFL introduces an Agent-Interpolation Initialization Module (AI2M), a new query initialization method for properly balancing foreground coverage and content learning.
AI2M integrates FPS with learnable queries to produce resilient queries.

Also, BFL proposes a Hierarchical Query Fusion Decoder (HQFD) to retain low overlap queries, mitigating the decrease in recall with the deepening of transformer layers.

Extensive experiments on benchmark datasets (ScanNetV2, ScanNet200, ScanNet++, and S3DIS) show that BFL performs superior 3D instance segmentation.

**Strengths:**

- The paper is fairly well-written and easy to follow.

- The authors consider a set of agents consisting of position and content queries to better initialize queries.
A discussion of existing query initialization methods (FPS-based and Learnable-based) is helpful for understanding the proposed approach with Agent interpolation.

- The experimental results look promising, and the proposed method, BFL, outperforms previous methods on various benchmark datasets.

**Weaknesses:**

- The motivation for the proposed method needs to be clarified.
The authors discuss the object disappearance phenomenon and limitations of multi-layer transformers using an example from a single scene in Figure 1.
Also, in Figure 2 (b), it is unclear how many scenes are included to calculate recall scores.
These examples seem insufficient to support the limitations of the multi-layer transformer, which has proven effective across various vision tasks.

- The authors mentioned that objects like picture and bookshelf are difficult to predict.
However, these objects achieved higher accuracy scores than others, like counter and window, as shown not only in Table 13 of the Appendix but also in various methods on the ScanNetV2 leaderboard.
This makes the definition of "difficult-to-predict" instances unclear.
It would be better to demonstrate that the proposed method has indeed improved the accuracy of these objects.

- It would be more reasonable to visually demonstrate (from a more detailed angle) whether the proposed method effectively resolves the object disappearance phenomenon, one of the motivations.

- In Table 6, adding a comparison of AP, precision, and recall metrics for the S3DIS dataset as other papers would be beneficial to validate the robustness of the approach.

**Questions:**

- The authors mentioned that noisy features lead to unstable directions in query optimization in line 313.
It would be helpful to provide a more detailed explanation of how these noisy features hinder query optimization.

- In HQFD, low overlap queries from the previous layer are concatenated with queries in the next layer.
Are these low overlap queries confident?
If low confidence queries accumulate, could this potentially lead to negative effects?

- In Table 8, when comparing the scores in the second row (S=400, L=400) with those in the fifth row (S=400, L=200) and the sixth row (S=200, L=400), it appears that the performance decline is more significant when varying the number of sampled points.
Could you explain why the number of sampled points (S) seems to have a greater impact on performance compared to the number of agents (L)?

While this work is well written, technical soundness is somewhat limited.
I have a few questions, as outlined in the weaknesses and questions.
A clarification of the points I mentioned would help me improve my decision.

---

### Official Review · Reviewer_68if · 2024-11-05

**Soundness:** 2
**Presentation:** 2
**Contribution:** 2
**Rating:** 3
**Confidence:** 5

**Summary:**

This paper proposes a 3D instance segmentation approach called BFL. It addresses challenges with query initialization and recall consistency in transformer-based methods. BFL introduces two main innovations: the Agent-Interpolation Initialization Module and the Hierarchical Query Fusion Decoder. AI2M combines position and content information through farthest point sampling (FPS) and learnable content queries, aiming to enhance coverage and content learning. HQFD mitigates the problem of inter-layer recall decline by retaining low-overlap queries, helping to maintain instance recognition as layers deepen.

**Strengths:**

1. Extensive testing on multiple benchmarks (e.g., ScanNetV2, ScanNet200).
2. The architecture and methodologies are explained in a structured manner, including ablation studies that assess the impact of different components.

**Weaknesses:**

1. While the paper introduces techniques to improve query initialization and recall, the results show only marginal gains over recent methods like Maft, particularly on common benchmarks in metrics like AP and mAP. Furthermore, models like Spherical Mask outperform BFL in terms of a range of metrics. Thus, it is difficult to perceive the proposed method as revolutionary.
2. The innovations in the paper are primarily focused on minor architectural adjustments (e.g., fusion techniques and query initialization schemes). Nowadays, these tricky designs will not provide significant new insights into 3D segmentation or address fundamental limitations in current transformer designs. For instance, can these designs significantly solve hard cases for 3D instance segmentation?
3. The paper reports an increase in runtime compared to MAFT, coupled with the limited performance gain, raises concerns regarding the model’s true insights.  As this method has heavy dependence on existing techniques, I just believe it's not interesting.

**Questions:**

See the Weaknesses.

---

### Official Review · Reviewer_nd4r · 2024-11-07

**Soundness:** 3
**Presentation:** 2
**Contribution:** 3
**Rating:** 8
**Confidence:** 4

**Summary:**

This work builds on top of a previous transformer-based 3D instance segmentation method (Maft), and suggests improvement in the query initialization and retention. For the query initialization, it proposes a combination between learnable and a non-learnable approach (FPS) using interpolation. For query retention, it proposes to keep queries at decoder layers where their correposnding masks do not overlap with the masks of the subsequent layer queries. Experiments on four common 3D instance segmentation datasets show improvement over the baseline.

**Strengths:**

- The proposed method focuses on the drawbacks of the baseline in terms of the recall across the decoder layer and the query initialization. The presented work is mainly additive to the baseline, with queries being intitialized with a combination between FPS and learned, and queries being added to the subsequent layers without replacing existing queries. This seems as a valid direction to improve upon the baseline.
- The experimental setup is well-structured and the diverse datasets (four datasets) strengthens the validity of the results.
- The ablation studies clearly show the importance/contribution of each of the proposed modules.

**Weaknesses:**

- The paper in some parts lacks clarity. For example, most of the paper contribution is in section 3.3.2, where more explanation would help understand the motivation of the choices made. This section is also not well connected to the main pipeline figure (Figure 3) as it does not mention FPS.
- The proposed additions require various hyperparameters (number of sampled points,number of agents, number of neighbours, NMS parameters, distance threshold, number of layers to retain queries from). While the proposed approach shows improved results on various datasets, each dataset required a different set of hyperparameters (appendix). This indicates an additional training time requirement to select the best set of hyperparameters.
- Test set results on ScanNet200 are missing.
- Paper organization and visuals can be improved. For example, discussions on Table 5 appear very early (L236). Some visuals are inconsistent: Figure 2 shows FPS after agent, Figure 3 shows it in parallel, text says FPS before agent (L207).

**Questions:**

- Some figures do not provide enough information, such as Figure 1 (how are objects extracted from those layers?)
- ScanNetv2 benchmark results do not appear on the leaderboard (it would be good to have it public to show more class-wise comparison across the different metrics).
- It is unclear what section 3.3.1 a and Table 1 are meant to convey. the FPS distance is dependent on the scene size, and the suggestion of sampling 100% of foreground distance is not data supported.
- L138: "It proves to be tailored for navigating complex and dynamic environments." How is this related to dynamic environments?
- What is the Zero in "FPS + Zero" "Learnable/Zero"
- Table 3 typo: column heading should be scannet++ instead of scannetv2
- While the increase in runtime is mentioned, could the authors provide more details on whether this includes the additional IoU computation, post-processing (NMS), and agent KNN computation?

---

### Official Review · Reviewer_Patb · 2024-11-10

**Soundness:** 3
**Presentation:** 3
**Contribution:** 3
**Rating:** 6
**Confidence:** 3

**Summary:**

The paper aims to address two primary limitations of existing transformer-based methods: (i) the difficulty in simultaneously maintaining strong positional and content information during query initialization, and (ii) a issue of object disappearance as decoder layers deepen, due to supervision at each layer. To tackle these issues, the authors introduce (a) an Agent-Interpolation Initialization Module, designed to create queries that achieve a balance between foreground positional coverage and content learning. (b) Additionally, they propose a Hierarchical Query Fusion Decoder that preserves low-overlap queries, mitigating the decrease in recall as layers deepen and thereby addressing the object disappearance problem. The methods are evaluated on the ScanNetV2, ScanNet200, ScanNet++, and S3DIS datasets.

**Strengths:**

1. The paper identifies two key issues with existing transformer-based approaches and proposes simple yet effective solutions.

2. Overall, the main ideas of the paper are easy to follow, although the clarity in the method section could be improved.

3. Experiments on diverse datasets demonstrate the merits of the proposed contributions over baselines and existing approaches.

**Weaknesses:**

1. Increase font size in Fig. 3 for readability—currently, the font size is too small.

2. Why is the proposed query initialization module named “Agent-Interpolation”? The term “agent” may misleadingly suggest autonomous agents. A more intuitive name could avoid confusion.

3. What motivates the use of “Bottom-K masks” for selecting low-overlapping masks? Why not using an IoU threshold (either learnable or fixed) which might be more intuitive for determining overlap.

4. Some bold claims, such as the method being “tailored for navigating complex and dynamic environments,” lack supporting experimental results or adequate explanations.

5. Several additional hyperparameters of the proposed approach—like the number of agents, distance threshold, and layers for query selection—must be individually tuned per dataset (as per the appendix). How consistent are results if these hyperparameters  are kept the same across datasets?

6. Test set results on ScanNet200 should ideally be included in the paper.

**Questions:**

1. How many masks from the previous layer are considered low-overlapping. i.e., what is K in Bottom-K?

2. In cases of mask disappearance, is there any possibility that certain masks have zero overlap in subsequent decoder masks? If so,  what happens if the IoU is zero for more than K masks—are all of them considered?

3. I strongly recommend submitting the ScanNetV2 results to the public leaderboard to facilitate more detailed comparisons across metrics.

4. It’s unclear what “Zero” refers to in “FPS + Zero”—does it mean FPS alone?

---

### Meta-Review · Area_Chair_YRcp · 2024-12-21

**Metareview:**

The manuscript received overall ratings of 3, 6, 3, and 8. While the reviewers appreciated that the manuscript is well-written with experiments on diverse datasets, they also raised several concerns including, missing hyperparameter analysis, test set results on ScanNet200, lack of experimental support behind some bold claims in the manuscript, marginal gains over recent methods like Maft, limited novelty (e.g, minor architectural adjustments), and increase in runtime compared to MAFT. Authors submitted the rebutall to address the concerns of the reviewers. Some of the concerns were addressed, such as results on ScanNet200 testset. However, two reviewers remained negative mentioning limited novelty (e.g., MAFT and other earlier methods have proposed strategies to enhance recall) and performance of the proposed approach against more recent methods on ScanNetV2. Reviewer's concerns also expressed that the rationale and compelling insights behind the methodological design remained unclear. Given the reviewers comments, rebuttal and discussions, the recommendation is reject.

**Additional Comments On Reviewer Discussion:**

While the reviewers appreciated that the manuscript is well-written with experiments on diverse datasets, they also raised several concerns including, missing hyperparameter analysis, test set results on ScanNet200, lack of experimental support behind some bold claims in the manuscript, marginal gains over recent methods like Maft, limited novelty (e.g, minor architectural adjustments), and increase in runtime compared to MAFT. Authors submitted the rebutall to address the concerns of the reviewers. Some of the concerns were addressed, such as results on ScanNet200 testset. However, two reviewers remained negative mentioning limited novelty (e.g., MAFT and other earlier methods have proposed strategies to enhance recall) and performance of the proposed approach against more recent methods on ScanNetV2. Reviewer's concerns also expressed that the rationale and compelling insights behind the methodological design remained unclear.

---

### Decision · Program_Chairs · 2025-01-22

Reject